# Universal Black-Box Reward Poisoning Attack Against Offline Reinforcement Learning

## Abstract

We study the problem of universal black-boxed reward poisoning attacks against general offline reinforcement learning with deep neural networks. We consider a black-box threat model where the attacker is entirely oblivious to the learning algorithm, and its budget is limited by constraining the amount of corruption at each data point and the total perturbation. We require the attack to be universally efficient against any efficient algorithms that might be used by the agent. We propose an attack strategy called the 'policy contrast attack.' The idea is to find low- and high-performing policies covered by the dataset and make them appear to be high- and low-performing to the agent, respectively. To the best of our knowledge, we propose the first universal black-box reward poisoning attack in the general offline RL setting. We provide theoretical insights on the attack design and empirically show that our attack is efficient against current state-of-the-art offline RL algorithms in different learning datasets.

## 1 Introduction

Reinforcement learning (RL) in the offline setting (Levine et al., 2020; Agarwal et al., 2020) has become a promising framework for realizing reinforcement learning applications in practice. Offline RL avoids the necessity of potentially expensive online data collection and can work with offline data directly. Taking medical treatment problems as an example, collecting online data would require testing on humans, which is dangerous and infeasible. In contrast, abundant recordings of how a human doctor treats a patient are available for offline training.

A number of effective algorithms have been proposed for the offline RL problem and achieve promising empirical performance on simulated environments. (Bhardwaj et al., 2024; Kumar et al., 2020; Fujimoto & Gu, 2021; Kidambi et al., 2020; Wu et al., 2019; Cheng et al., 2022). However, a practical threat arises in the real world that is usually overlooked in the simulations. In many applications, the offline data is usually based on human feedback (Zheng et al., 2018; Kiran et al., 2021), making it possible for an adversary to poison the reward signal in the training dataset. For example, the learning agent may ask a third-party agent to provide offline data. If the third-party agent is dishonest, it may provide human data with subtle malicious changes that are hard to detect but can potentially cause unwanted results for the learning agent. The threat of poisoning attacks even draws attention in recent reinforcement learning from human feedback (RLHF) Baumgärtner et al. (2024); Wang et al. (2023); Wu et al. (2024), and people are concerned that such attacks may greatly influence the performance of current RLHF approaches. The state-of-the-art offline reinforcement learning algorithms mentioned above only consider ideal environments without such threats. A few works Li et al. (2024); Ye et al. (2023) investigate the settings with the poisoning attack. However, they only consider the attacker adopting some random attack strategies, which may not be enough to evaluate the robustness of an algorithm properly. Therefore, it is important to investigate the vulnerabilities of current algorithms to the poisoning attacks with more efficient attack strategies.

In Fig 1, we show the framework of the attack. Specifically, we focus on a universal black-box reward-poisoning attack with a limited budget such that under the attack, any originally efficient learning algorithm can no longer learn a high-performing policy. Such an attack setting is more realistic due to the following reasons:

1. **Universal and Black box**: First, it is not in the interest of the learning agent to reveal its algorithm to the attacker. Second, offline RL algorithms are very different in learning strategies

and frameworks. Therefore, it is hard for the attacker to assume which kind of learning algorithm the agent will use. The attacker has to work without any knowledge of the algorithm used by the agent and expect the agent to choose any efficient algorithm arbitrarily.

2. **Reward poisoning**: In RL environments, the reward signals are usually provided by the environment, such as human users, while the state signals are determined by the agent. For example, in reinforcement learning from human feedback (RLHF), the state signals are the prompts the learner gives, while the reward signals are feedback from humans. Therefore, it is more feasible for the attacker to poison rewards.

3. **Limited budget**: Too much perturbation on the reward signals makes it easier for the learning agent to detect. So, it is more practical for the attacker to work with limited budgets and perturb the training process as little as possible.

We formally introduce our criteria for an attack to be practical in Section 3.

**Challenges:** We find the following challenges for developing a practical reward poisoning attack.

1. **Fixed training dataset:** In the online learning setting, an attacker can mislead the learner to collect data that only covers states and actions with low long-term rewards so that the learner never observes how a high-performing policy would work, making it fail to learn a high-performing policy (Xu et al., 2022). In contrast, the training dataset is fixed in the offline learning setting. This results in the famous 'pessimistic' learning (Levine et al., 2020) in offline RL, and the learning agent is inclined to stick to the behaviors covered by the dataset. The learner can always observe how an expert would behave (the action) in the dataset regardless of the attack. This contributes to the fundamental difficulty of misleading the learning agent.

2. **Universal and Black box attack:** A practical attacker should be oblivious to the learning agent and universal efficiency against any algorithms it might use. First, it is challenging to list all possible efficient learning algorithms that an agent might use. Second, without knowing the details of an offline learning algorithm, it would be challenging to predict the exact learning result of the algorithm under the attack.

**Contributions:** To address the challenges above, our insight is to characterize general efficient offline RL algorithms with the pessimistic learning feature. For a pessimistic algorithm, it will find nearly the best policy well covered by the dataset. Based on the insight, we construct an attack framework called the 'adversarial reward engineering framework' such that one can evaluate the efficiency of an attack against universal efficient algorithms in the framework. To find an efficient attack under the framework, the key idea of our attack is to find the high-performing policies covered by the dataset and make them appear low-performing to the agent and vice versa. In this case, the algorithm will believe that some poor-performing policies are near optimal. We build our main attack method, 'policy contrast attack,' based on the idea and theoretically analyze the insight behind it based on general assumptions. To the best of our knowledge, this is the first universal black-box reward poisoning attack in the general offline RL setting. We empirically test the efficiency of our attack on various standard datasets from the D4RL benchmark (Fu et al., 2020) learned by different state-of-the-art offline RL algorithms (Kumar et al., 2020; Fujimoto & Gu, 2021; Kostrikov et al., 2021). Our attack is efficient in most cases with a limited attack budget. We also show that our attack is not sensitive to the choice of hyper-parameters. One can use our attack to assess the practical robustness of offline RL algorithms. We hope that our work can inspire the development of more robust offline RL algorithms.

## 2 RELATED WORK

**Reward poisoning attack in online RL:** Most studies on reward poisoning attacks against RL focus on the online learning setting. Zhang et al. (2020a); Huang et al. (2017) investigate observation perturbation attack where the attacker modifies the state signals to adversarial examples during training. Rakhsha et al. (2020); Zhang et al. (2020b) theoretically investigate the attack problem in the online RL setting with tabular MDPs. They show that given a specific learning algorithm or an algorithm with certain learning guarantees, attack strategies exist such that the learning algorithm will learn a low-performing policy even if the attack only slightly perturbed the training process. However, many of these are white-box attacks, and it remains unknown how to scale these techniques

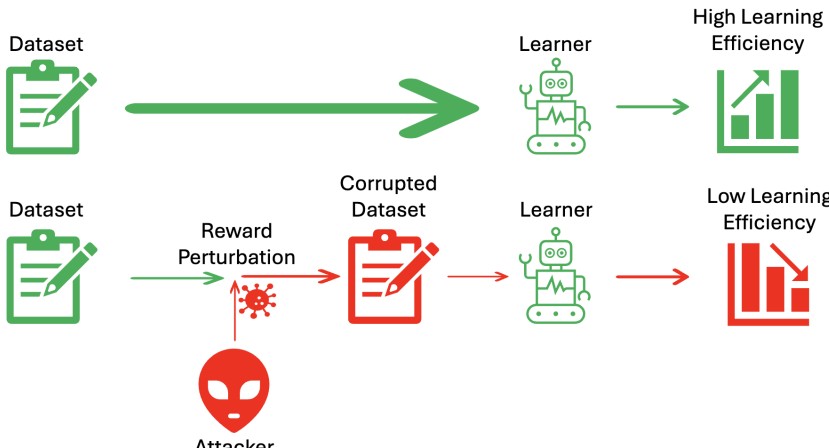

Figure 1: Reward poisoning attack framework.

to general RL environments that require function approximation with deep neural networks. In the more complicated general RL setting, Sun et al. (2020) empirically shows that current state-of-the-art DRL algorithms are vulnerable under the poisoning attack when the attacker is aware of the learning algorithm. Xu et al. (2022); Xu & Singh (2023) further shows that even in the black-box setting where the attacker is oblivious to both the environment and the learning algorithm, the data poisoning attack can achieve targeted and untargeted attack goals with limited budgets.

**Reward poisoning attack and defense in offline RL:** There is limited work on the poisoning attack in offline RL. Ma et al. (2019) theoretically studies the problem for tabular MDPs that cannot scale to the more complicated general RL setting considered in our work. Furthermore, their attack assumes that the learning agent can learn the globally optimal policy, which is not appropriate and practical for recent offline RL algorithms. Li et al. (2024) theoretically and empirically shows that the learning efficiency of an efficient offline RL algorithm is related to the dataset quality regardless of the reward signals, which is closely related to our assumption in Section 4. To defend against the attack, Zhang et al. (2022); Ye et al. (2023) design algorithms and theoretically prove their robustness. They also empirically test their algorithm against some specific random attack strategies. We empirically show that our attack is much more efficient than such random attacks for different learning scenarios. No attack exists for general offline RL except for the random attacks. Still, we also adapt a state-of-the-art online RL attack from Xu et al. (2022) to our setting as an additional baseline.

## 3 PRELIMINARIES

**Offline RL:** We consider a standard offline RL setting (Cheng et al., 2022) where an agent trains on an offline dataset to learn how to perform well in an RL environment. An RL environment is characterized by an MDP $\mathcal{M} = (\mathcal{S}, \mathcal{A}, \mathcal{P}, \mathcal{R})$ where $\mathcal{S}$ is the state space, $\mathcal{A}$ is the action space, $\mathcal{P} = \mathcal{S} \times \mathcal{A} \to \Delta(\mathcal{S})$ is the state transition function, and $\mathcal{R} = \mathcal{S} \times \mathcal{A} \to \mathbb{R}$ is the reward function. A policy $\pi : \mathcal{S} \to \Delta(\mathcal{A})$ is a mapping from the state space to a distribution over action space suggesting the way one behaves in the environment. Without loss of generality, we assume the initial state is always $s_0$. Then the performance of a policy $\pi$ for the environment is defined as $J_{\mathcal{R}}(\pi) = \mathbb{E}_{\pi, \mathcal{P}}[\sum_{t=0}^{\infty} \gamma^t \mathcal{R}(s_t, a_t) | a_t \sim \pi(s_t)]$, which represents the long-term discounted cumulative rewards for running the policy in the environment.

An offline reinforcement learning dataset is a collection of observation tuples collected from the environment: $\mathcal{D} = \{(s_i, a_i, s_i', r_i), i = [0, \ldots, N]\}$ where $s_i' \sim \mathcal{P}(s_i, a_i)$ and $r_i \sim \mathcal{R}(s_i, a_i)$. The state action distribution $\mu = \{(s_i, a_i), i = [0, \ldots, N]\}$ represents the trajectories covered in the dataset. An offline reinforcement learning agent has access to the offline dataset $\mathcal{D}$ as well as the state and action spaces $\mathcal{S}, \mathcal{A}$, and its goal is to find a high-performing policy.

**Reward poisoning attack against offline RL:** We consider a reward poisoning attack model where a malicious adversary can poison the reward signals in the offline dataset to mislead the learning

agent. Formally, for the $i^{\text{th}}$ observation tuple $(s_i, a_i, s_i', r_i)$, the attacker can inject a perturbation $\Delta_i$ to the reward signal, then the corrupted observation tuple becomes $(s_i, a_i, s_i', r_i + \Delta_i)$. We denote $\boldsymbol{\Delta} = \Delta_1, \ldots, \Delta_N$ to be the corruption strategy of the attack. For a practical threat model, the attacker should have limited abilities. Formally, we consider the following constraints on the attacker.

1. **No access to the offline training process:** Since the training is offline, the learning agent can host the training process on local machines. Therefore, it would be impractical for the attacker to observe the training process of the learning agent.

2. **No knowledge of the learning algorithm:** There are a variety of efficient offline RL algorithms with very different learning strategies. Unless the learning agent announces its learning algorithm to the public, it would be impractical for the attacker to assume which learning algorithm the agent would choose. Therefore, the attacker should not have any detailed knowledge of the learning algorithm.

3. **Limited Budget:** To make the attack stealthy, the attacker should make the corruption as small as possible. First, the attacker should limit the maximal perturbation on each reward signal $||\boldsymbol{\Delta}||_\infty = \max |\Delta^i|$, as an extreme value of reward would make the data point very suspicious. Second, the attacker should limit the total amount of perturbation $||\boldsymbol{\Delta}||_1 = \sum_{i=0}^{N} |\Delta^i|$. The reason is that too many slightly problematic data points can also make the agent realize that the dataset is unreliable.

Given a fixed budget, the general goal of the attacker is to make the learning agent learn a low-performing policy. We will formally formulate the attacker's goal as an optimization problem in the next section.

## 4 POLICY CONTRAST ATTACK

In this section, we formally define and formulate the universal black-box offline RL attack as an optimization problem. We consider the attacks that are universally effective against any data-driven learning algorithms of high learning efficiency. We highlight the challenges in finding the efficiency of an arbitrary attack and focus on a framework called the 'adversarial reward engineering framework' where the behavior of any efficient algorithm becomes predictable under any instance of the framework. Finally, we present our attack algorithm, 'policy contrast attack,' which is an efficient instance under the framework.

### 4.1 UNIVERSAL ATTACK ON GENERAL OFFLINE REINFORCEMENT LEARNING AS AN OPTIMIZATION PROBLEM

Here, we formally formulate the universal black-box offline RL attack as an optimization problem. Let $\mathcal{L}$ be a class of offline RL algorithms that a learning agent might use. Let $\pi^{\text{Alg}}(\mathcal{D} \oplus \boldsymbol{\Delta})$ be a random variable representing the policy learned by a learning algorithm Alg training on the dataset $\mathcal{D}$ with corruption $\boldsymbol{\Delta}$. By applying a limited amount of perturbation, the attacker should make the agent learn a low-performing policy no matter which algorithm it chooses from $\mathcal{L}$.

$$
\begin{aligned}
\min_{\Delta_{1:N}} \ & V \\
\text{s.t.,} \ & \forall \text{Alg} \in \mathcal{L}, \mathbb{E}[\mathcal{J}_{\mathcal{R}}(\pi^{\text{Alg}}(\mathcal{D} \oplus \boldsymbol{\Delta}))] \leq V \\
& ||\boldsymbol{\Delta}||_0 \leq B, ||\boldsymbol{\Delta}||_1 \leq C
\end{aligned}
\tag{1}
$$

The problem is not complete yet as we have not specified $\mathcal{L}$, the class of algorithms an agent might use. The problem becomes trivial if $\mathcal{L}$ includes all possible policies. In this case, the agent can choose an algorithm that outputs a specific policy with maximal performance regardless of the reward signals poisoned by the attack. Then, all attacks that satisfy the budget constraints are 'optimal' solutions. As a result, it is meaningless for the attacker to consider all possible algorithms, especially the ones that are not reward-driven and inefficient. A rational learning agent will only consider using an algorithm that 'efficiently' learns the reward signals to find a decent policy based on the dataset. Therefore, it is more important for an attack to focus on the effect of the attack against such algorithms. For

this purpose, in this work, we assume $\mathcal{L}$ to include all $\delta$-optimal pessimistic learning algorithms in Assumption 4.1 below.

**Assumption 4.1.** ($\delta$-optimal pessimistic learning algorithms assumption) Given an offline RL dataset $\mathcal{D}$ collected from an environment $\mathcal{M} = (\mathcal{S}, \mathcal{A}, \mathcal{P}, \mathcal{R})$ with state-action distribution $\mu$. For all efficient offline RL algorithm $\text{Alg} \in \mathcal{L}$ that might be used by the agent, there exists a small value $\delta > 0$ and a class of policies $\Pi_\mu$ supported by the distribution $\mu$ such that $J_\mathcal{R}(\pi^{\text{Alg}}(\mathcal{D})) \geq \max_{\pi \in \Pi_\mu} J_\mathcal{R}(\pi) - \delta$. That is, the agent will always use an algorithm that can learn the policy with nearly the highest performance on $\mathcal{M}$ among the supported policies.

Here, we justify that Assumption 4.1 is practical and can represent current efficient offline RL algorithms. It has become common sense that pessimistic learning is the key to efficiency in offline RL Levine et al. (2020). A pessimistic learning algorithm avoids the policies not supported by the dataset to minimize the chance of reward hacking, that is, outputting a policy that appears promising by the dataset but has poor performance. Li et al. (2024) has the same insight in explaining the 'survival instinct' of existing offline RL algorithms against attack: a pessimistic algorithm will stick to the policies covered by the dataset so that it won't output any low-performing and out-of-distribution policies regardless of the reward signals.

By the optimization problem in Eq 1, we can naturally define the efficiency of an attack as follows.

**Definition 4.2.** $((V, B, C)$-efficient attack) An attack is $(V, B, C)$-efficient if it satisfies the following conditions:

$$\forall \text{Alg} \in \mathcal{L}, (\mathcal{D} \oplus \boldsymbol{\Delta})) \leq V$$
$$\|\boldsymbol{\Delta}\|_0 \leq B, \|\boldsymbol{\Delta}\|_1 \leq C \tag{2}$$

Given attack budgets $B$ and $C$, finding the optimal attack for Eq 1 is equivalent to finding the most efficient $(V, B, C)$-efficient attack with the minimal value of $V$. Given an arbitrary attack strategy, the attack budgets $B$ and $C$ can be computed by the attack construction. However, evaluating the learning outcome of the agent under the attack is challenging. First, the influence of a universal attack should be evaluated on all algorithms from $\mathcal{L}$, and it is challenging to find all the algorithms that satisfy the near-optimal efficient learning conditions. Second, evaluating the exact influence of an attack on a specific learning algorithm is challenging in the black-box setting, as the neural network used by the agent is unknown to the attacker. In conclusion, it is challenging to evaluate the exact efficiency of an arbitrary attack; hence, it is also challenging to find the optimal attack. To deal with the challenges and find an attack of high efficiency, we adopt an attack framework where one can analyze the efficiency of the attack instances under the framework.

## 4.2 Adversarial Reward Engineering Framework

**Definition 4.3.** (Adversarial reward engineering framework) In the adversarial reward engineering framework, an instance of attack constructs an adversarial reward function $\widehat{\mathcal{R}}$. The corresponding corruption on the $i^{\text{th}}$ data $(s_i, a_i, r_i)$ from $\mathcal{D}$ satisfies $\Delta_i = \widehat{\mathcal{R}}(s_i, a_i) - r_i$.

In the adversarial reward engineering framework, it is equivalent to say the corrupted dataset is collected from the adversarial environment $\widehat{\mathcal{M}} = (\mathcal{S}, \mathcal{A}, \mathcal{P}, \widehat{\mathcal{R}})$ with the same state-action distribution $\mu$ as the original dataset. Therefore, we can apply assumption 4.1 to characterize the behavior of the agent under the attack: it will learn a supported policy that is near-optimal on the adversarial reward function $\widehat{\mathcal{R}}$. Formally, the efficiency of an instance of adversarial reward engineering framework is guaranteed in Lemma 4.4 below.

**Theorem 4.4.** *Let $\widehat{\mathcal{R}}$ be the adversarial reward of an instance of the adversarial reward engineering framework. Let $\widehat{\Pi}^* := \{\pi | \pi \in \Pi_\mu, J_{\widehat{R}}(\hat{\pi}^*) \geq \max_{\pi \in \Pi_\mu} J_{\widehat{R}}(\pi) - \delta\}$ be the $\delta$-optimal supported policies in $\widehat{\mathcal{R}}$. The efficiency of the attack satisfies $V \leq \max_{\pi \in \hat{\pi}^*} J_\mathcal{R}(\pi)$, $B = \max_i |\widehat{\mathcal{R}}(s_i, a_i) - r_i|$ and $C = \sum_i |\widehat{\mathcal{R}}(s_i, a_i) - r_i|$.*

The proof for all theorems and lemmas can be found in the Appendix. Recent works about robust offline RL widely consider an attack we call 'random inverted reward attack' for empirical evaluation against their learning algorithms Zhang et al. (2022); Ye et al. (2023); Li et al. (2024). Such an attack randomly flips the signs of a portion of rewards in the datasets and can be treated as an instance of

the adversarial reward engineering framework. One can find a detailed analysis in the Appendix. Next, we show a way to construct a more efficient instance of the adversarial reward engineering framework and justify the design of our policy contrast attack.

### 4.3 Towards Efficient Adversarial Reward Engineering Attack: Policy Contrast Attack (PCA)

To ensure that the agent learns a low-performing policy, an efficient instance of the adversarial reward engineering attack needs to make the optimal or near-optimal supported policies in the adversarial environment have poor performance in the true environment. So, our goal is to find a way to construct such an adversarial reward model while limiting the required attack budget. However, it is still impractical to find all near-optimal supported policies and their performance in the true environment as the environment dynamics are unknown to the attacker. Therefore, we focus on finding a feasible way to construct a reward function such that any near-policy policies in the reward function are likely to perform poorly in the true environment.

In Theorem 4.5, we show a sufficient condition and a necessary condition for the adversarial reward function to make the agent learn a low-performing policy with performance less than $V$. The two conditions are the same when $\delta = 0$.

**Theorem 4.5.** *Consider an attack in the adversarial reward engineering framework with $\widehat{\mathcal{R}}$. To make the actual performance of the learned policy by an arbitrary $\delta$-optimal pessimistic learning algorithm lower than a value $J_{\mathcal{R}}(\pi_0) < V$, a sufficient condition is that the adversarial reward function satisfies*

$$\exists \pi_1 \in \Pi_\mu.\ J_{\mathcal{R}}(\pi_1) < V, \forall \pi_2 \in \Pi_\mu. J_{\mathcal{R}}(\pi_2) \geq V,$$
$$J_{\widehat{\mathcal{R}}}(\pi_1) > J_{\widehat{\mathcal{R}}}(\pi_2) + \delta,$$

*and a necessary condition that it should satisfy is*

$$\exists \pi_1 \in \Pi_\mu.\ J_{\mathcal{R}}(\pi_1) < V, \forall \pi_2 \in \Pi_\mu. J_{\mathcal{R}}(\pi_2) \geq V,$$
$$J_{\widehat{\mathcal{R}}}(\pi_1) > J_{\widehat{\mathcal{R}}}(\pi_2) - \delta,$$

*In both cases, we call $\pi_1$ that satisfies the conditions 'bad policy' and $\pi_2$ 'good policy.'*

Theorem 4.5 suggests that the attacker should make a low-performing policy have a higher performance in the adversarial reward function than any high-performing policies. Formally, denote $\Delta J(\pi) = J_{\widehat{\mathcal{R}}}(\pi) - J_{\mathcal{R}}(\pi)$, to make $J_{\widehat{\mathcal{R}}}(\pi_1) > J_{\widehat{\mathcal{R}}}(\pi_2) + \delta$, the attacker needs to ensure that $\Delta J(\pi_1) - \Delta J(\pi_2) > J(\pi_2) - J(\pi_1) + \delta$. In other words, the attacker needs to make $\Delta J(\pi_1)$ high for some bad policy $\pi_1$ and $\Delta J(\pi_2)$ low for any good policy $\pi_2$. To limit the cost of the attack, the attacker should also determine the rewards to poison that have more influence on $\Delta J(\pi_1)$ and $\Delta J(\pi_2)$. This directly inspires the key idea for finding an efficient attack that in the adversarial reward function, the rewards for some low-performing policies are increased so that they appear as high-performing and vice versa.

In terms of finding the bad policies, we show in Lemma B.2 that the attacker can find nearly the worst policy supported by the dataset by training on the dataset of all rewards inverted.

**Lemma 4.6.** *When training on the dataset $\mathcal{D}$ with the signs of all rewards inverted, the learning agent will learn the policy supported by the dataset with nearly the worst performance $V \leq J_{\mathcal{R}}(\pi_0) = \min_{\pi \in \Pi_\mu} J_{\mathcal{R}}(\pi) + \delta$.*

In terms of finding the good policies, Lemma 4.7 shows that policies of high performance will behave similarly to some policies from the good policy set constructed through Alg 2.

**Lemma 4.7.** *Let $V$ be the performance of the learned good policy in the last iteration of Alg 2. For any policy $\pi \in \Pi_\mu$ such that $\min_{s \in \mathcal{S}, \pi_2 \in \Pi_2} d_a(\pi(s), \pi_2(s)) > d$, its performance satisfies $J_{\mathcal{R}}(\pi) < V + \delta$.*

After finding the good and bad policies, the final step is to make them look bad and good in the adversarial environment. This can be achieved straightforwardly by decreasing or increasing the rewards associated with the actions given by the good or bad policies.

---

**Algorithm 1** Policy Contrast Attack

---
**Input:** dataset $\mathcal{D}$ of size $N$, efficient offline RL algorithm Alg
**Params:** distance measure $d_a$, distance threshold $d$, corruption parameters $\Delta_1$, $\Delta_2$, number of iterations $K$
Initialize $\widehat{\mathcal{D}} = \{\}$.
Learn a bad policy $\pi_1$ from dataset with inverted rewards $-\mathcal{D}$
Learn a good policy set $\Pi_2$ through Alg 2 with $K$ iterations
**for** $i = 1$ **to** $N$ **do**
    Get state-action-reward $(s_i, a_i, r_i)$ from $\mathcal{D}$
    **if** $\min_{\pi \in \Pi_2} d_a(a_i, \pi(s_i)) \le d$ **then**
        Modify $\hat{r}_i = -\Delta_2$
    **end if**
    **if** $d_a(a_i, \pi_1(s_i)) \le d$ **then**
        Modify $\hat{r}_i = r_i + \Delta_1 \cdot (1 - d_a(a_i, \pi_1(s_i)/d)$
    **end if**
    Update $\widehat{\mathcal{D}} = \widehat{\mathcal{D}} \cup \{(s_i, a_i, \hat{r}_i)\}$.
**end for**
**Output:** $\widehat{\mathcal{D}}$

---

Based on the discussion above, we formulate 'policy contrast attack.' The attack consists of two parts as follow:

1. **Bad policies look good:** The attacker trains on the training dataset with all rewards inverted and learns 'bad' policies. Then, the attacker poisons the dataset by increasing the rewards associated with the actions close to that given by the 'bad' policies.

2. **Good policies look bad:** The attacker iteratively learns a set of 'good' policies with Algorithm 2. Then, the attacker poisons the dataset by decreasing the rewards associated with the actions close to that given by the 'good' policies.

Formally, the policy contrast attack is given in Alg 1. Here $\Delta_1$, $\Delta_2$, and $d$ are the hyper-parameters of the attack and set to utilize given attack budgets $B$ and $C$ fully. $d_a$ is a measure of the distance between actions. For the actions from a continuous space, we choose $d_a(a_1, a_2) = ||a_1 - a_2||_2$. The corresponding adversarial reward function in Alg 1 satisfies:

$$
\widehat{\mathcal{R}}(s, a) = \begin{cases} \mathcal{R}(s, a) + \Delta_1 \cdot (1 - \frac{d_a(a, \pi_1(s))}{d}), & \text{if } d_a(a, \pi_1(s_2)) \le d \\ -\Delta_2, & \text{else if } \min_{\pi_2 \in \Pi_2} d_a(a, \pi_2(s)) \le d \\ \mathcal{R}(s, a), & \text{otherwise} \end{cases}
$$

The first part of the reward function is to increase the rewards associated with the bad actions given by the bad policy. The formation of the reward modification in this part follows Xu & Singh (2023). The penalty on the reward is a linear function depending on the distance between an action and the bad policy's action. This is to make it easier for the agent to converge to the bad policy. The second part of the reward function is to decrease the rewards associated with good policies. The penalty here is simply making the reward a low value, as in this case, we merely want the agent to learn something different from the good policies.

Note that there are some cases where the actions given by the good and bad policies are close. There will be a conflict for the aforementioned corruption strategy in this case. Intuitively, these cases are less common as usually good and bad actions should not be similar. In addition, in these cases, a change in reward has a similar influence on the performances of good and bad policies in the adversarial function, suggesting that the corruption in these cases has a limited impact on the ranking of good and bad policies in the adversarial reward. To ensure that the bad policy always performs better in the adversarial environment, we always increase the reward associated with the bad actions whenever there is a conflict.

| D4RL Dataset | Normal | Fully inverted | PCA | Rand 30% | RPP | RPI |
|---|---|---|---|---|---|---|
| HalfCheetah-ME - TD3_BC | $91.0 \pm 2.9$ | $13.1 \pm 1.9$ | $\mathbf{76.2 \pm 2.1}$ | $89.8 \pm 3.2$ | $88.95 \pm 2.65$ | $81.3 \pm 5.1$ |
| Hopper-ME - TD3_BC | $105.9 \pm 3.3$ | $16.9 \pm 6.3$ | $\mathbf{82.1 \pm 6.4}$ | $99.8 \pm 8.9$ | $88.22 \pm 8.37$ | $97.4 \pm 6.9$ |
| HalfCheetah-MR - IQL | $44.3 \pm 0.4$ | $1.07 \pm 0.1$ | $37.8 \pm 3.2$ | $43.5 \pm 0.8$ | $\mathbf{1.72 \pm 0.98}$ | $43.9 \pm 1.1$ |
| Walker2D-M - CQL | $81.8 \pm 0.7$ | $67.7 \pm 2.0$ | $\mathbf{73.1 \pm 0.2}$ | $77.4 \pm 2.1$ | $77.45 \pm 1.66$ | $80.1 \pm 0.4$ |
| Walker2D-MR - IQL | $80.2 \pm 3.6$ | $-0.2 \pm 0.1$ | $\mathbf{31.9 \pm 1.5}$ | $78.7 \pm 5.5$ | $60.21 \pm 3.8$ | $37.5 \pm 2.8$ |
| Hopper-MR - CQL | $85.2 \pm 8.1$ | $0.7 \pm 0.01$ | $30.3 \pm 4.7$ | $25.1 \pm 4.9$ | $86.05 \pm 13.64$ | $\mathbf{19.2 \pm 10.3}$ |
| Sum Totals | $488.4$ | $99.27$ | $\mathbf{331.4}$ | $414.3$ | $402.6$ | $359.4$ |

Table 1: Performance of learning algorithms under the attacks.

## 5 EXPERIMENTS

### 5.1 PERFORMANCE OF DIFFERENT ATTACK STRATEGY

We test the effectiveness of our attack against current state-of-the-art offline RL algorithms training on the standard D4RL offline RL benchmarkFu et al. (2020). We show that with limited attack budgets, our policy contrast attack is generally more effective.

**Experiment setup:** The offline datasets we choose from the D4RL benchmark include different combinations of environments and types of trajectories. We consider HalfCheetah, Hopper, Walker2d, and Ant environments. We consider the data collected by a single medium policy (medium), a single expert policy (expert), a single random policy (random), a mixture of medium and expert policies (medium-expert), and the replay buffer of a medium policy (medium-replay).

For the learning algorithms, we choose IQL  Kostrikov et al. (2021), CQL Kumar et al. (2020), and TD3-BC Fujimoto & Gu (2021), which are frequently used as baselines in current offline RL studies. These algorithms are implemented based on CORL library Tarasov et al. (2022). Recall that the attacker also needs a learning algorithm to learn good and bad policies. To ensure that the attacker has no information about the learning agent, the learning agent and the attacker always use different learning algorithms.

We choose three baseline attacks. The first one is the random invert attack, the only non-trivial attack against offline RL in the current literature. The second and the third ones are the random policy invert (RPI) attack and random policy promote (RPP) attack proposed in Xu et al. (2022) for the online setting, as it is the only existing attack closely related to and can be adapted to work in the offline setting. In the appendix, we show the result of using only one part of the attack strategy from the policy contrast attack.

For the attack budget, following Ye et al. (2023), we always set the budget for $B$ and $C$ to be the amount required by the random inverted reward attack that corrupts $30\%$ of the data. In the ablation study, we tested the influence of different attack budgets on the attack's efficiency.

For the hyper-parameters of the attack, the corruption parameters $\Delta_1$ and $\Delta_2$ are set to fully utilize the budget $B$, and the attack radius $d$ is set to fully use the budget $C$. The number of good policies, i.e., the number of iterations $K$, is set as 5. In the appendix, we show that the algorithm's efficiency is not sensitive to the choice of corruption parameters and the number of good policies. We run 5 times for each experiment with different random seeds and show the average result.

Note that the construction of our attack does not depend on the assumption we made for the learning algorithms. Although the learning algorithms may not strictly satisfy the assumption of efficient learning algorithms, we show that our attack is still effective.

**Main Results:** The offline RL algorithms we choose iteratively update their learned policies. To straightforwardly show the effectiveness of attacks, we use the performance of the policy learned by the agent at the end of training to represent its learning efficiency. In the appendix, to intuitively show the details of the training process under the attacks, we show the performance of the policies learned by the algorithm at each iteration during training. We will also show the full training log for the experiments in the ablation studies.

To cover as diverse offline datasets as possible, here we use only one learning algorithm for each dataset. Next, we will show the case where we test the performance of different learning algorithms on the same dataset poisoned by our attack. Table 1 shows the main results of running our policy

contrast attack in different learning scenarios. The naive baseline is the training with no attack. The ideal effect of the attack is the training with fully inverted rewards, as the learning algorithm tries to learn the worst-performing policy in this case. We observe that our policy contrast attack is usually effective for any combination of datasets and learning algorithms. In contrast, the random inverted attack has almost no effect on the learning result except for the Hopper-Medium Replay dataset, where both attacks are similarly effective. In many cases, our attack is also significantly more efficient than the RPI and RPP baseline attacks.

Here, we briefly discuss why the policy contrast attack should be more efficient than other baseline attacks. 1. **random inverted attack:** At a high level, both the policy contrast and the inverted reward attack want to inverse the agent's ranking of good and bad policies. The inverted reward attack treats all state actions equally and inverse the performance of all policies. In contrast, the policy contrast attack focuses on the policy and the more important state actions. This could be the reason for making the policy contrast attack more efficient in terms of budget. 2. **RPI and RPP attacks:** The PCA attack focuses on the supported policies, while the RPI and RPP attacks focus on arbitrary policies. Since the efficient learning algorithm mostly considers only supported policies, the poisoning of such policies should have more influence on the learning process.

## 5.2 UNIVERSAL ATTACK AND ROBUSTNESS EVALUATION

To verify that our attack is universal against different learning algorithms, we take the Hopper-medium-expert dataset as an example, apply the policy contrast attack using the same setup as in the main results, and run four different learning algorithms including TD3 BC, CQL, AWAC, and ReBRAC on the corrupted dataset. The results in Figure 5.2 show that our attack is always effective against all four learning algorithms.

In addition, an important direct application of our attack strategy is using the attack to evaluate the robustness of an algorithm. In this example, we find that compared to other algorithms, the AWAC algorithm performs much better under our attack, suggesting that it is more robust.

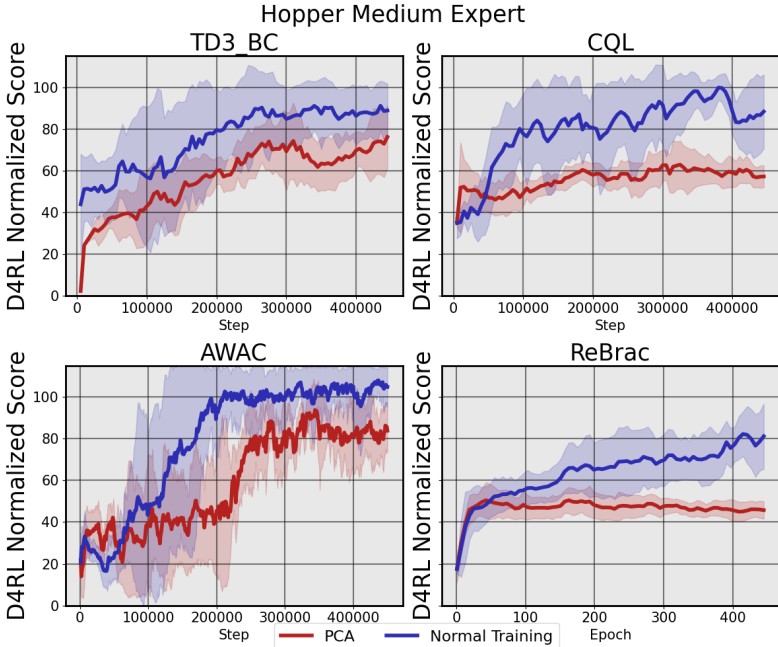

Figure 2: Peformance of different learning algorithms on the same dataset under the attacks.

## 5.3 INFLUENCE OF ATTACK BUDGET

**Attack budget $B$:** Here we study the influence of per-step attack budget $B$ on the attack efficiency. Denote $r_{\max} = \max_i |r_i|$ as the maximal absolute reward from the dataset. In the main results, the

budget required by the attacks is $B = 2 * r_{max}$. Here, we set the value of $B$ to be $1.5 \cdot r_{max}$ and $1 \cdot r_{max}$ and test on two randomly chosen datasets. The results in Fig 3 show that our policy contrast attack remains efficient with a lower budget of $B$. We note that the attack is slightly more efficient with less value of $B$. Intuitively, although the per-step corruption is less, the attack influences the performance of more policies, making it possible for the agent to learn slightly worse policies. The result also suggests that the efficiency of our attack is not significantly affected by the choice of the corruption parameters $\Delta_1$ and $\Delta_2$ at each step, as their values depend on $B$.

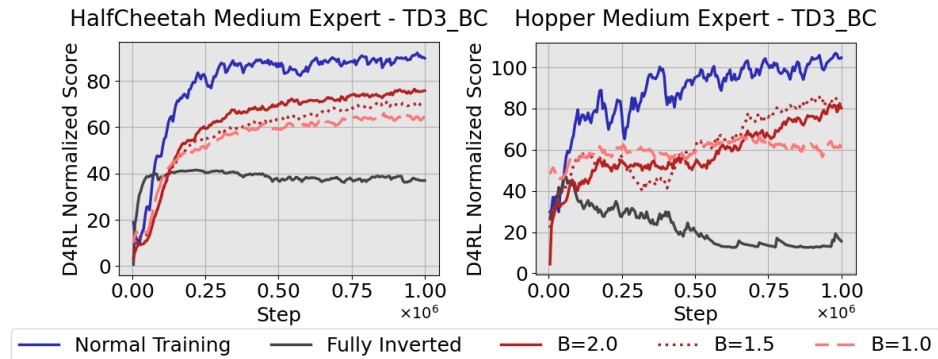

Figure 3: Influence of different $B$ budget on the attack.

**Attack budget $C$:** Here, we study the influence of attack budget $C$, the total amount of corruption, on the efficiency of the attack. We set the value of $C$ to be the amount required by a random inverted attack that corrupts $20\%$ and $40\%$ of data. The results in Fig 4 show that our attack is more efficient with a higher budget of $C$ and remains effective with a lower budget of $C$.

We also study the effect of other hyperparameters on our method. We find that the size of the good policy set has a very small influence on its efficiency. If our method only has the part of making good policy look bad or making bad policy look good, it will be much less efficient than the full method. The details of empirical results can be found in the Appendix.

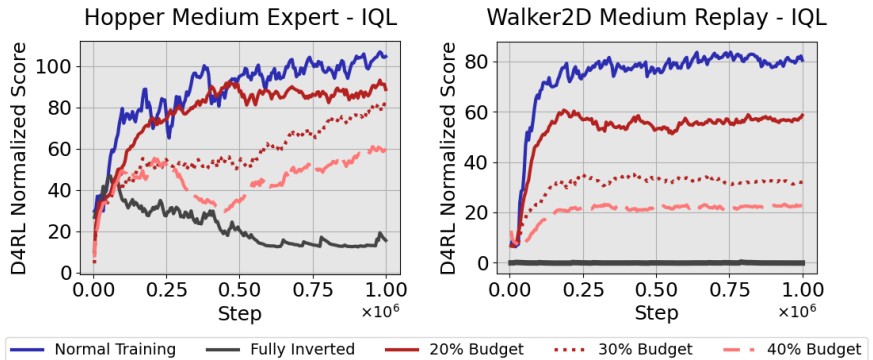

Figure 4: Influence of different $C$ budget on the attack.

## 6 CONCLUSION AND LIMITATION

In this work, we propose the first black-box reward poisoning attack in offline reinforcement learning based on our novel theoretical insights. Our empirical results suggest that the attack is effective in various learning scenarios. However, our work is limited in that we only consider black-box attacks, and the attacker requires access to the full offline training dataset.

## 7 REPRODUCIBILITY

In the main paper, we explain the setting of the problem we study and the threat model we consider. The proofs for all theorems and lemmas can be found in the appendix. The codes we use for the experiments can be found in the supplementary materials.

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

## A    PROOF FOR THEOREMS AND LEMMAS

**Proof for Theorem 4.4** By the definition of the attack, the perturbation on the $i^{\text{th}}$ data is $\Delta_i = \widehat{\mathcal{R}}(s_i, a_i) - r_i$. By Assumption 4.1, the agent should learn a near-optimal policy on the reward function of the dataset, which is $\widehat{\mathcal{R}}$ under the attack. Therefore, the policy $\hat{\pi}^*$ learned by the agent under the attack satisfies $J_{\widehat{R}}(\hat{\pi}^*) \geq \max_{\pi \in \Pi_\mu} J_{\widehat{\mathcal{R}}}(\pi) - \delta$.

**Proof for Theorem B.2** The adversarial reward function constructed by the fully inverted attack is $\widehat{\mathcal{R}} = -\mathcal{R}$. By Assumption 4.1, the agent will learn a policy $\pi_0$ that is near-optimal on $\widehat{\mathcal{R}}$: $J_{\widehat{\mathcal{R}}}(\pi_0) \geq \max_{\pi \in \Pi_\mu} J_{\widehat{\mathcal{R}}}(\pi) - \delta$. Therefore, the performance of the learned policy on the true reward function satisfies $J_{\mathcal{R}}(\pi_0) = -J_{\widehat{\mathcal{R}}}(\pi) \leq -(\max_{\pi \in \Pi_\mu} J_{\widehat{\mathcal{R}}}(\pi) - \delta) = \min_{\pi \in \Pi_\mu} J_{\mathcal{R}}(\pi) + \delta$.

**Proof for Theorem 4.5** First, we prove the sufficient condition. For any policy $\pi_2 \in \Pi_\mu : J_{\mathcal{R}}(\pi_2) > V$, we have $\max_{\pi \in \Pi_\mu} J_{\widehat{\mathcal{R}}}(\pi) \geq J_{\widehat{\mathcal{R}}}(\pi) \geq J_{\widehat{\mathcal{R}}}(\pi_2) + \delta$. Therefore, $\pi_2$ is not a near-optimal policy on the adversarial reward function, and by Assumption 4.1, the agent will not learn such a policy.

Next, we prove the necessary condition. Let $\pi_0$ be the policy learned by the agent under the attack, and this policy has a performance less than $V$ on the actual reward function $J_{\mathcal{R}}(\pi_0) < V$. Therefore, we can take $\pi_1 = \pi_0$. By Assumption 4.1, the gap between the performance of $\pi_1$ and any other policy on $\widehat{\mathcal{R}}$ must be less than $\delta$. Therefore, for any policy $\pi_2 \in \Pi_\mu : J_{\mathcal{R}}(\pi_2) > V$, we have $J_{\widehat{\mathcal{R}}}(\pi_1) > J_{\widehat{\mathcal{R}}}(\pi_2) - \delta$.

**Proof for Theorem 4.7** At the last iteration of Alg 2, the performance of the policy $\pi_K$ learned in the iteration on the adversarial reward function $\widehat{\mathcal{R}}_K$ in the iteration is no greater than $V$ because the adversarial reward function is strictly less than the actual reward function. By Assumption 4.1, the performance of any policy $\pi \in \Pi_\mu$ on $\widehat{\mathcal{R}}_K$ satisfies $J_{\widehat{\mathcal{R}}_K}(\pi) \leq V + \delta$. For any policy $\pi \in \Pi_\mu : \min_{\pi_2 \in \Pi_2, s} ||\pi(s) - \pi_2(s)||_2 > d$, their performance on the actual reward function and the adversarial reward function are the same. Therefore, their actual performance must be less than $V + \delta$.

## B    INVERTED REWARD ATTACK

Here, we discuss the most popular offline reward poisoning attack widely considered in recent RL studies Ye et al. (2023); Zhang et al. (2022); Li et al. (2024). We start from an illustrative example first to help better understanding the attack. Following the idea of reward engineering, one can immediately find an attack that can make the agent learn the worst possible policy under Assumption 4.1.

**Definition B.1** (Fully inverted reward attack). The fully inverted reward attack sets $\widehat{\mathcal{R}} = -\mathcal{R}$ as the adversarial reward function where $\mathcal{R}$ is the true reward function of the underlying environment behind the dataset. The corresponding attack strategy of the fully inverted reward attack satisfies $\Delta_i = -2 \cdot r_i$.

The inverted reward attack flips the sign of all the rewards in the dataset. In Theorem B.2 we show the efficiency of the fully inverted reward attack.

**Theorem B.2.** *Under the fully inverted reward attack, the learning agent will learn the policy supported by the dataset with nearly the worst performance $V \leq J_{\mathcal{R}}(\pi_0) = \min_{\pi \in \Pi_\mu} J_{\mathcal{R}}(\pi) + \delta$. The budgets required by the attack are $B = 2 * \max_i |r_i|$ and $C = 2 * \sum_i |r_i|$.*

Theorem B.2 can be derived from Theorem 4.4 directly. Maximizing the cumulative reward on $-\mathcal{R}$ is equivalent to minimizing that on $\mathcal{R}$. Therefore, an efficient learning algorithm will learn the policy with the worst performance on the true reward function among the policies supported by the dataset. Despite achieving the worst outcome for the learner, the inverted reward attack faces the problem of requiring high budgets, which is impractical. Since a practical attack should only corrupt a small portion of data, we consider an attack that can only randomly invert a part of the reward. We call this attack the 'random inverted reward attack'. This attack strategy has been widely used in previous works as a non-trivial attack Ye et al. (2023); Zhang et al. (2022).

**Definition B.3** (Random inverted reward attack). The random inverted reward attack has a parameter $p \in (0, 1)$. The attack randomly samples $p \cdot N$ indexes from $[1, \dots, N]$ with replacement, where $N$

is the size of the dataset. Let the set of indexes be $I$. The attack strategy of the inverted reward attack satisfies $\Delta_{i \in I} = -2 \times r_i$ and $\Delta_{i \notin I} = 0$.

We cannot find any strong guarantee for the effect of the attack on the learning agent. Empirically, we observe that when the attack inverts $< 50\%$ of the states, the learner is usually still able to learn a policy almost as good as the one it learns in the uncorrupted dataset. Since the random inverted reward attack is not practical, we need to find a more efficient attack that can work with less budget yet still makes the agent learn a policy of low performance.

The inverted reward attack treats different state actions from the dataset equally, yet the reward for some state actions can have more influence on the learning process. Therefore, it is possible to construct more efficient attacks that focus the attack budget on the rewards in some specific state actions.

## C  TRAINING LOG OF MAIN RESULTS

Here, we show the performance of the learned policies during training in different datasets under different attacks.

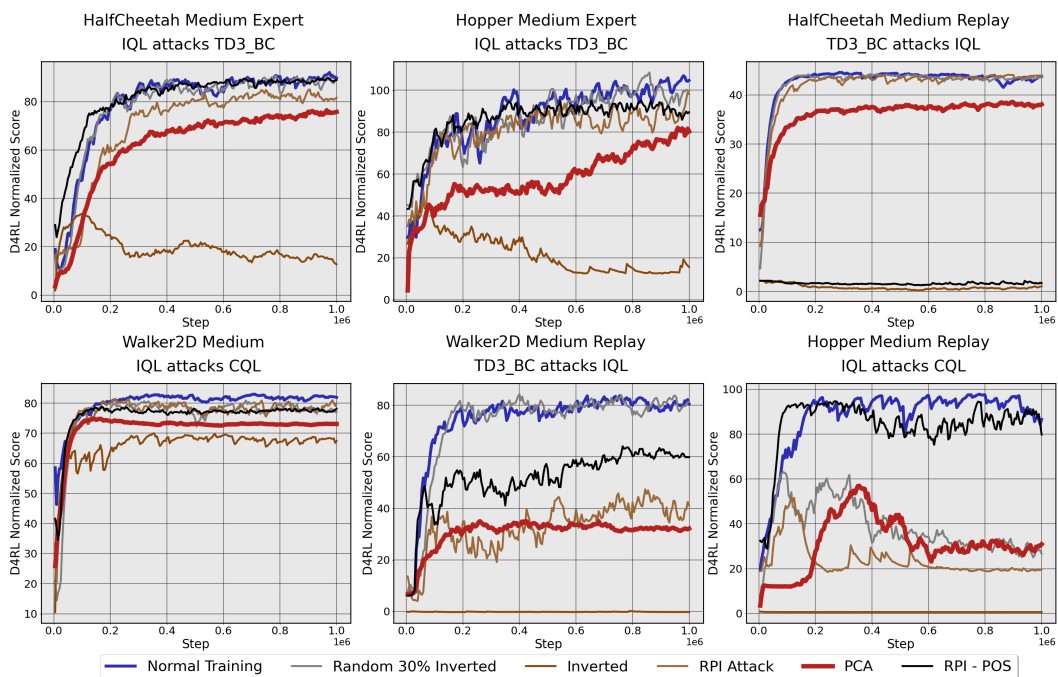

Figure 5: Peformance of learning algorithms on different datasets under the attacks. We take the title of the first figure as an example to explain the meaning of the title. 'HalfCheetah' means the RL environment is HalfCheetah; 'Medium Expert' means the dataset is collected by a mixture of medium and expert policies. 'IQL attacks TD3_BC' means the learning algorithm used by the attacker is IQL, and the one used by the learning agent is TD3_BC.

## D  ALGORITHM FOR LEARNING A SET OF GOOD POLICIES

## E  ADDITIONAL ABLATION STUDY

**Size of good policy set:** Here, we study the influence of the size of $\Pi_2$ on the efficiency of the attack. We set the sizes of the good policy set to be $3, 5, 7, 10, 15$. In Fig 6, we observe that the learning outcomes of the algorithms are similar for different sizes of good policy sets. The results suggest that the number of good policies does not considerably affect our policy contrast attack.

**Algorithm 2** Learning Set of Good Policies

**Input:** dataset $\mathcal{D}$ of size $N$, number of iterations $K$, offline RL algorithm Alg
**Params:** distance threshold $d$, corruption $\Delta$
Initialize $\Pi_2 = \emptyset$, $\widehat{\mathcal{D}} = \mathcal{D}$.
**for** $i = 1$ **to** $K$ **do**
    Learn $\pi$ from dataset $\widehat{\mathcal{D}}$
    Update $\Pi_2 = \Pi_2 \cup \{\pi\}$
    **for** $j = 1$ **to** $N$ **do**
        Get state-action-reward $(s_i, a_i, r_i)$ from $\widehat{\mathcal{D}}$
        **if** $\min_{\pi \in \Pi_2} d_a(a_i, \pi(s)) \leq d$ **then**
            Modify $r_i = -\Delta$
        **end if**
        Update $(s_i, a_i, r_i)$ in $\widehat{\mathcal{D}}$
    **end for**
**end for**
**Output:** $\Pi_2$

**Only Good/Bad Policies attacks:** To show that it is beneficial to do both making good policies look bad and bad policies look good, we test the efficiency of the attack strategy with only one part. That is, the attack either only decreases the reward associated with the good policies or increases the reward associated with the bad policy. In Fig 7, we observe that both attacks lose some efficiency compared to the policy contrast attack. Here, we provide empirical insights on why these two attacks do not work alone. For making good policies look bad, it can happen in practice that the good policy set does not cover all policies of high performance, therefore the agent is still able to identify a high-performing policy. For making a bad policy look good, in practice, even if the bad policy already has the highest performance in the adversarial environment, without the good policy attack part, the learning agent may not be able to learn it and still converge to a good policy as the adversarial reward function is more complicated. In this case, breaking the optimality of the good policies makes it easier for the learning agent to converge to the bad policy.

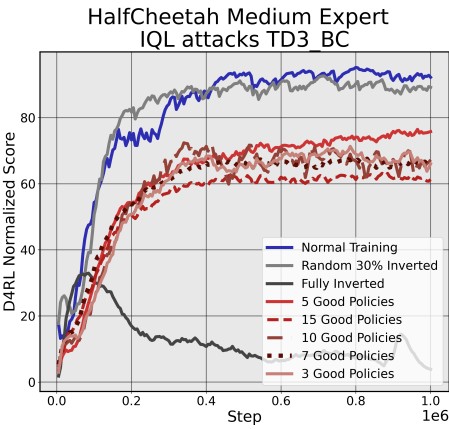

Figure 6: Influence of different sizes of good polices on the attack.

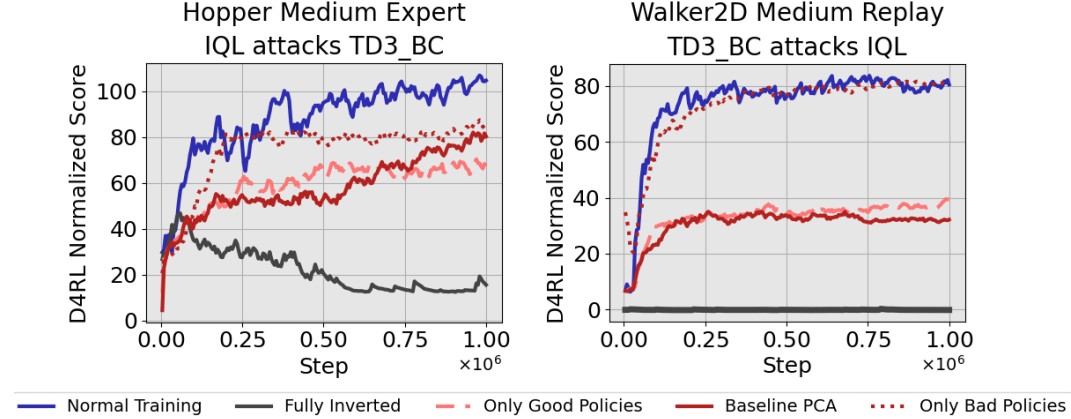

Figure 7: Attacks that only make bad policies look good and good policies look bad.

