# OpenReview forum: "Universal Black-Box Reward Poisoning Attack against Offline Reinforcement Learning"
_ICLR.cc/2025/Conference — ICLR 2025 Conference Withdrawn Submission_

### Official Review · Reviewer_PzkE · 2024-10-17

**Soundness:** 1
**Presentation:** 2
**Contribution:** 2
**Rating:** 3
**Confidence:** 5

**Summary:**

The authors claim a new, universal reward poisoning attack against offline reinforcement learning algorithms. Their attack assumes no prior information regarding the chosen algorithm of the training, victim party - instead aiming to generate reward perturbations that will be effective against any algorithm. They claim the first, universal poisoning attack against offline reinforcement learning in this black box setting.

They frame their objective as a constrained optimization problem aiming to find reward perturbations that are most likely to minimize the value of policies found by $\delta$-pessimistic offline RL algorithms. The adversary is restricted in the $l_\infty$ and $l_1$ norm of their reward perturbations.

They aim to achieve this objective with two main steps. They first learn a policy set $\Pi_2$ given the dataset $\mathcal{D}$ and then a sub-optimal policy $\pi_1$ given inverted rewards $\hat{r}_i = -r$. Using these policies they then decrease the rewards of "good" actions in the dataset $\mathcal{D}$ which are closer to $\Pi_2$ and increase the reward of "bad" actions close to $\pi_1$ in $\mathcal{D}$. The goal of this is that the victim agent should learn that "good" actions (according to the benign, unpoisoned MDP) are actually bad, while "bad" actions are actually good. Their attack is constrained by budget hyper-parameters $\Delta_1$ and $\Delta_2$.

They evaluate their attack on a suite of D4RL benchmarks and show they can decrease the expected return of the agent a decent amount more than more naive attacks under their reward poisoning constraints.

**Strengths:**

The paper has a few strengths
* Researching poisoning attacks against RL, and especially offline RL, is an important direction of research, so I appreciate the authors for aiming to add to the body of literature on this topic.
* The study is motivated within the context of most related work.
* The paper is fairly easy to follow.
* Their attack improves over their proposed baselines.

**Weaknesses:**

**Motivation -** The paper's motivation is placed fairly well within the context of related works, but I don't think the attack's objectives are well motivated within the context of a real world attack. This attack formulation is indiscriminate, meaning the agent's performance in all states is harmed. Lets now assume the attack is perfectly successful, minimizing $J_\mathcal{R}(\pi)$. Given this outcome, the party training the agent will be able to observe that the agent is completely failing to solve the desired task, so they'll never actually deploy it in the real world. This essentially makes the offline dataset $\mathcal{D}$ useless after poisoning, which could be one angle for why this attack formulation is interesting, but the trained agent will never actually cause negative outcomes down the line. For this same reason I think the slightly weaker objective of decreasing $J_\mathcal{R}(\pi)$ below some $V$ is also poorly motivated.

I think the authors should better motivate their attack objectives in the context of something a real adversary would hope to induce. I further suggest they compare and justify these objectives against those of backdoor poisoning attacks [1,2,3] which are much stealthier. It may also be interesting if the authors shift their adversarial objective to be less indiscriminate, perhaps only affecting a subset of trajectories or states rather than all states.

**Theory -** The paper's theoretical results are disconnected from the method that is developed and the claims they make in text. To me Theorem 4.4 is the main theoretical claim of the paper (though this is not completely clear based on the writing), which seems to claim that $J_\mathcal{R}(\pi) \geq V$ given some $\pi$ trained on poisoned rewards $\hat{\mathcal{R}}$. This seems to go against their objective of $J_\mathcal{R}(\pi) < V$. The result doesn't make sense to me and isn't supported by the proof given in the appendix. The proof only claims that the agent will learn a $\delta$-pessimistic policy over $\hat{\mathcal{R}}$, which I agree with, but it doesn't show how this relates to the value of the policy ($V$) under $\mathcal{R}$. Due to this I don't think the claim made in theorem 4.4 is correct. Furthermore, even if Theorem 4.4 is true, the theorem does not connect back to the method presented in algorithm 1, meaning there is nothing meaningful to say about how we should expect the attack to perform. Theorem 4.5 is also a bit confusing since it seems to be just presenting the attack's objectives rather than any guarantees afforded by the attack.

The paper also claims "universal" poisoning attacks, but only considers $\delta$-pessimistic algorithms in their theory. I don't think their method necessarily relies on targeting $\delta$-pessimistic algorithms (again since the theory doesn't seem to connect to the method), but I still think making the claim of "universal" is too strong without better theoretical results. I suggest the authors consider the effect that their poisoning has on the optimal policy of the MDP rather than performing their analysis over the class of learning algorithms. See [4] for examples of such analysis.

The proofs found in the appendix are also very short and informal, so I am not completely convinced by them. I think they make intuitive sense, but they rely really heavily on Assumption 4.1 doing the heavy lifting.

**Presentation -** the paper has a lot of notation and grammar mistakes. In particular the text claims the attacker wants to minimize the $l_\infty$ norm of their attack, which makes sense, but equations 1 and 2 consider the $l_0$ norm. I think if the authors intended to use $l_\infty$ in these equations, but I could be wrong. Also see line 282 "near-policy policies".

Another issue is that the paper frequently overloads terms and notation. For instance, in the text under equation 1 the authors seem to refer to $\mathcal{L}$ as both a set of policies and a set of algorithms. After reading the paragraph a few times  I believe I understand what the authors are saying, but I think even this adds to the confusion. I agree that one cannot consider all possible algorithms, as I can always design an algorithm that outputs an arbitrary policy, but I don't think these cases are important to the paper's analysis. Instead I recommend the authors just consider the affect their attack has over the optimal policy given the dataset rather than considering a set of algorithms.

There are also a few pieces of the paper which I think are redundant and can be removed to save space. For instance, equations 1 and 2 are nearly identical, along with the two conditions of theorem 4.5. The fact that the attack is black box is stated 3 times in bullet points under "Universal and Black Box", "Universal and Black box attack", and "No knowledge of the learning algorithm".

**Experimental Results -** The paper's experimental results are okay, but they aren't too strong. I think a more novel attack formulation would be required to have stronger results under these constraints. An attack objective which only considers a subset of states may be able to perform better.

**Summary -** I think the study of indiscriminate attacks against offline RL isn't well motivated since they just result in policies that are never deployed. I would be okay with this attack formulation if the theoretical or empirical results were novel, but I think this paper needs stronger theoretical analysis and a better methodology before it can meet that criteria.

[1] "BAFFLE: Hiding Backdoors in Offline Reinforcement Learning Datasets" https://arxiv.org/abs/2210.04688

[2] "TrojDRL: Trojan Attacks on Deep Reinforcement Learning Agents" https://arxiv.org/abs/1903.06638

[3] "SleeperNets: Universal Backdoor Poisoning Attacks Against Reinforcement Learning Agents" https://arxiv.org/abs/2405.20539

[4] "Understanding the Limits of Poisoning Attacks in Episodic Reinforcement Learning" https://arxiv.org/abs/2208.13663

**Questions:**

* Can you please explain the results you claim with theorem 4.4? It is hard to understand what the theorem is really stating.
* Why should an attacker implement this form of attack over a backdoor attack, given they have access to the data?

---

### Official Review · Reviewer_rsaN · 2024-10-28

**Soundness:** 3
**Presentation:** 2
**Contribution:** 3
**Rating:** 5
**Confidence:** 3

**Summary:**

In this study, the authors studied and develop the idea of developing an efficient black-box reward poisoning attack on efficient offline RL algorithms with the aim of the attack model being universally applicable to all efficient offline RL algorithms. Specifically, the authors proposed two algorithms, one for learning the good policies and one for learning the bad policies, and use swap these two policies to augment the rewards in the offline RL dataset in order to fool to offline RL agent during training. The authors showed that their methods are more effective and efficient compared to baseline methods on several MuJoCo environments and also provided theoretical motivations and backings for their formulation.

**Strengths:**

The paper presents a theoretically motivated idea, applying a well-known concept (altering rewards by flipping the signs of good and bad policies) from adversarial attacks to the offline reinforcement learning (RL) setting. The approach is straightforward and repurposes existing ideas in a new context, broadening the potential applications of adversarial methods. The paper is generally easy to follow, with a clear structure that helps convey the main concepts and findings, making it accessible. While the results are only compared with very simple baselines, they offer initial empirical insights into how this approach might perform within offline RL and suggest possible directions for further exploration.

**Weaknesses:**

The paper’s contribution would be strengthened by a deeper, more comprehensive empirical foundation, as many of the current results are sampled from a subset of possible experiments, limiting the generalizability of the claims. The absence of certain details, such as the definitions of variables and terms early in the text, hinders the flow and clarity of the presentation. Additionally, including additional details on the baselines used as comparison would improve the reader's understanding of the novelty and effectiveness of the proposed method.  Addressing these elements would help establish the reliability and broader applicability of the findings.

**Questions:**

1. I understand the motivation of the authors for claiming this to be "universal", however, in my opinion, given that experimental results only show a only a limited subset of combination of offline RL algorithms x environments, claiming this method to be universal seems a little too strong.

2. Under the challenges section, the authors claim that one of the main challenge is that in offline RL, the agent is learning from a fixed dataset. "The learner can always observe how an expert would behave (the action) in the dataset regardless of the attack. This contributes to the fundamental difficulty of misleading the learning agent." Given that the fundamental idea of the paper is to present the high performing policy as a poor one and vice-versa in the dataset, is this truly a challenge ? I would suggest rephrasing this section as it seems like a trivial challenge that was highlighted specifically just to be solved by the proposed method.

3. In the Related Works section, cited works such as Zhang et al. (2020a); Huang et al. (2017) are more related to state perturbations, rather than reward poisoning.

4. In the Related Works section, the authors claim "However, many of these are white-box attacks, and it remains unknown how to scale these techniques to general RL environments that require function approximation with deep neural networks." Could the authors explain in more detail what do they mean by "scaling" and what challenges do they foresee that prevents the "scaling" of existing methods?

5. In general, what are the challenges of applying existing methods of reward poisoning methods in the online setting directly to offline setting? This needs to be succinctly clarified since the authors have repeatedly claimed that existing methods are not applicable/scalable.

6. In the case of the budget on each step of the reward signal, how will the existing algorithm work if the dataset collected consisted of a sparse reward trajectory where most of the rewards are 0? Would this still maintain the undetectability of the attack?

7. In Section 4.1, variables V, B and C are not introduced until a later section. These should ideally be introduced earlier to improve the readability. Similarly, in Theorem 4.5, $\pi_{1}$ and $\pi_{2}$ were not formally defined until later

8. I do not understand how this inequality $J_{R}(\pi_{\text{Alg}}(D)) \geq \max_{\pi \in \Pi_{\mu}} J_{R}(\pi) - \delta$ implies the agent will always use an algorithm that can learn the policy with nearly the highest performance among the supported policies. Isn't this inequality merely claiming that the offline RL (LHS) is greater than the online RL (RHS) - $\delta$, which really leaves the offline RL unbounded in terms of performance. How does that relate to the statement?

9. The baselines RPI and RPP should be discussed in a little more detail for the sake of completeness, given that they're the main source of comparisons in this paper.

10. Could the authors explain how are the combination of algorithm-environment selected? Are they randomly selected?

11. "In many cases, our attack is also significantly more efficient than the RPI and RPP baseline attacks." How are the authors defining efficiency here and where is this statement being supported?

12. In section 5.3, the authors state:"Intuitively, although the per-step corruption is less, the attack influences the performance of more policies, making it possible for the agent to learn slightly worse policies." What do you mean by the attack influences the performance of more policies? Also, how is the initial baseline of B = 2*$r_{max}$ selected? Furthermore, the authors claim that the attack is slightly more efficient with lower values of $B$, which seems counterintuitive. So, what is the main takeway of this section?

**Details Of Ethics Concerns:**

This paper proposes a method to poison a dataset in order to adversarially affect the training of offline RL algorithms, but the ethical implications of the proposed method was not explicitly discussed.

---

### Official Review · Reviewer_19yU · 2024-10-31

**Soundness:** 1
**Presentation:** 1
**Contribution:** 1
**Rating:** 3
**Confidence:** 5

**Summary:**

This paper proposes a universal black-box poisoning attack named "policy contrast attack" against offline reinforcement learning. The authors validate its effectiveness in various datasets from the D4RL benchmark.

**Strengths:**

- The data security of the offline RL datasets has a significant impact on the RL agent training.

- It is an interesting idea to design a poisoned reward mechanism to obfuscate the actual policy performance.

**Weaknesses:**

1. The presentation of this paper has a significant issue. Lots of expression in the manuscript are not academically professional. For example, "it is equivalent to say", "Here, ...". Some arguments seem to be overclaimed such as " theoretically analyze the insight behind it
based on general assumptions". I do not see any rigorous theoretical analysis when going through the main text and the appendix. In addition, the meanings of the variables in the math formulas are unclear. I suggest adding "where XXX is ..." after each equation.

2. I am confused about the inconsistency between the attack model (limited budget with the infinity-norm constraint) and the mathematical formulation (zero-norm constraint in Equation (1) and (2)). I assume the authors may want to choose the inifinity-norm?

3. Although the authors did an experiment to vary the corrupted data ratio, it seems the objective does not pose a constraint over the corrupted data ratio. I am wondering if it is necessary to incorporate such a constraint into the final objective. It would be great if the authors can have some discussion.

4. Regarding Theorem 4.4, I do not see any relationship between the efficiency of the attack and the inequalities of  B, V, C. I would appreciate it if the authors could further explain the relationship.

5. Regarding Theorem 4,5, first, it is unclear to me what $\pi_0$ is in the context. Second, the necessary condition gives a very loose condition over the poisoned performance over policy $\pi_1$ and $\pi_2$. When $\delta > 0$, the inequality even cannot ensure the poisoned performance of the "bad policy" is no worse than the "good policy". Third, given the original performance of policy $\pi_1$ and $\pi_2$, it is apparent that $\Delta J(\pi_1)$ should be much higher than $\Delta J(\pi_2)$. I do not think a full paragraph (line 302-line 309) is needed to claim this simple argument.

6. Due to the confusion of Theorem 4.5, Lemma 4.6 and Lemma 4.7 are challenging for me to understand. It would be great if the authors could provide further clarification.

7. For Line 319, $\pi(s)$ outputs an action only when it is a deterministic policy. Otherwise, the output should be a distribution. Therefore, $d_a$ is expected to be measuring the difference between two distributions, e.g., kl-divergence.

8. The compared baseline Xu et al. (2022) is relatively weak. More recent baselines (which are accepted by major ML/RL venues) are expected to be included for comparison. Otherwise, it is difficult to examine the effectiveness of the proposed attack.

9. I took a look over Ye et al. (2023), the authors varied the corruption ratio from 10% to 30% in Appendix D. I am wondering how the proposed attack performs under the ratio of 10%. In reality, corrupting too much data is not realistic and can be detected with high probability.

10. The conclusion for Figure 2 does not seem to be valid. Given the limitation of the original RL algorithm, the difference between the reward under normal training and the reward under PCA attack should be considered. In this view, TD3 algorithm is also robust.

**Questions:**

Please see the weaknesses above.

**Details Of Ethics Concerns:**

This work proposes a poisoning attack against offline reinforcement learning which may cause an ethical issue if using such a poisoned data to obtain a RL agent.

---

### Official Review · Reviewer_cLva · 2024-11-01

**Soundness:** 2
**Presentation:** 2
**Contribution:** 2
**Rating:** 3
**Confidence:** 4

**Summary:**

This paper approaches the problem of attacks in offline RL with quite strong constraints on what the adversary has access to. Without knowing the victim algorithm type, training process, or oracle set of solutions, the proposed approach learns an efficient adversarial reward function that inverses good and bad policy performance. Experiments show the success and efficiency of the method and compare it to similar attacks, even though true 1:1 comparisons don't yet exist.

**Strengths:**

The paper outlines a difficult problem and provides a fundamentally motivated and novel solution with solid inspiration from existing theoretical methods.

The intuition behind the approach is also clear and easy to understand.

The constraints on the problem and the importance of addressing them are well-stated.

**Weaknesses:**

The terms and notation used in the theoretical contributions are confusing. To list a few:
- $V$ is undefined in eq 1, it is normally a value function but does not appear to be so here.
- in assumption 4.1, $J_R (\pi^{alg}(D))$ and $J_R (\pi)$ are both used. Does the RHS of that equation use a different dataset?
- equations and definitions such as the $\delta$-optimal pessimistic learning algorithm, Thm 4.5, and $\hat(R)$ on page 7 should be clearly labeled and numbered to communicate how they are used.
- In Def 4.2, are there theoretical determinations for B and C? or, are they hyperparameters? The difference seems important to the understanding of "efficiency".

The motivation and definitions of the Policy Contrastive Attack seem to contain some self-defeating arguments:  In Challenges, it is stated that listing all efficient learning algorithms is problematic. However, the method relies on the ability to find "good" policies as outlined in Alg 2. This outlines questions about the feasibility of the method.

The authors state that only one learning algorithm for each dataset is shown to increase experimental diversity. However, there is no justification for the choices of each algorithm/dataset combination. It would be helpful to include all pairwise combinations in the appendix.

Tables and Figures:
-In black and white, both lines in Figure 2 look the same. A more descriptive caption would aid understanding.
-All tables and figures lack descriptive captions.

**Questions:**

1) What happens when $J_{\hat{R}}(\cdot) = 0$ for all inputs? The necessary condition will always be satisfied in Thm 4.5, but it seems unlikely such a reward function would cause the victim to specifically learn bad policies and not good ones. Are there theoretical limits to the formulation of $J_{\hat{R}}$?

2) Could the authors clarify the nature of the constraint $C$? In Figure 4, it is represented as a percentage, but in Definition 4.2 it functions as a value.

3) What makes the method "universal"? The distinction is unclear from reading the related works and section 5.2.

4) Would the method be feasible if instead of learning "good" policies in Algorithm 2, only "bad" ones were learned, and then the rewards inversed? It would be interesting to know, and if true would reduce concerns about the feasibility of learning all good policies.

---

### Official Review · Reviewer_UP62 · 2024-11-02

**Soundness:** 3
**Presentation:** 4
**Contribution:** 3
**Rating:** 6
**Confidence:** 4

**Summary:**

This paper proposes a black-box reward poisoning method for offline reinforcement learning that can be applied to any efficient offline RL algorithm. The authors first establish theoretical framework on the attack problem, then show sufficient condition and necessary condition of the attacker. Motivated by the conditions and analysis, the paper proposes a Policy Constrast Attack, which learns a bad policy and a good policy set, perturbing rewards such that good policy obtains lower reward and bad policy obtains high reward. Experiments show that the proposed method is effective on D4RL dataset and several different algorithms.

**Strengths:**

1. The paper is very well-written. The studied problem is clearly stated and motivated.
2. The theoretical analysis is helpful for understanding the essense and challenges of the reward poisoning problem.
3. The proposed algorithm makes intuitive sense.

**Weaknesses:**

The proposed algorithm is an approximated solution to the reward poisoning problem. But it is not clear how far it is to the optimal solution. Given the experiment results, there seems to be a big gap between the fully inverted attack and PCA attack. Is it possible to show some analysis?

**Questions:**

- In line 334, should it be $\hat{r}_i = \hat{r}_i - \Delta_2$?
- How to determine the hyperparameters $d$, $\Delta_1$, $\Delta_2$?

---

### Official Review · Reviewer_1QJW · 2024-11-04

**Soundness:** 3
**Presentation:** 2
**Contribution:** 2
**Rating:** 5
**Confidence:** 3

**Summary:**

This work presents a universal blackbox reward poisoning attack for offline RL. The proposed approach extends [ref1] to an offline setting, while providing additional evidence on the universality of the attack from various aspects such as learning algorithm, budget and other hyperparameters used in their approach. The provided ablations demonstrates the attack is effective across multiple environments and offlineRL policies including BC, CQL, and IQL.









[ref1] Xu, Yinglun, and Gagandeep Singh. "Black-Box Targeted Reward Poisoning Attack Against Online Deep Reinforcement Learning." arXiv preprint arXiv:2305.10681 (2023).

**Strengths:**

- the proposed approach provides a general framework for conducting blackbox reward poisoning attacks, and the reported results demonstrates reasonable performance on limited attack budget
- the general idea is simple and effective: using a set of learnt best and worst policies, and by leveraging a distance on actions each perform w.r.t the sampled action, identify how to corrupt a sampled reward from the dataset.

**Weaknesses:**

- In the main results in Table1, it is unclear what learning algorithm is used to learn best/worst policy leveraged by the attacker. The relation between learning algs used for best or worst policies by the attacker, and the attacked policy are also not studied. Please share this information and discuss the relationship.

- The evaluation framework lacks clarity: looking at the results (e.g, in Table1), it is not immediately clear whether PCA is best performing or not. E.g, total performance of fully inverted is the least, hence, should be considered the best method? why is PCA is better than fully inverted? Is the difference between PCA and RPI significant? what difference is significant?

- Perhaps a better budget would be to take into account number of poisoned samples, instead of budget. This measure is more universal and in some practical cases may be even more realistic. It is not clear why the budgets introduced are of importance, it seems more like these are constraints of the proposed method and are not universal. This part of evaluation, too, lacks further support and motivation, and needs to be improved by more general and practical constraints. Please provide discussions on the choices of constraints, and provide empirical evidence based on #samples as constraints.

- RL methods are very sensitive to hparam selection. If a non-optimal policy is used to adjust the reward, instead of an adversarial policy, how would such a baseline perform in comparison? How can one shows an adversarial attack as proposed is absolutely necessary to achieve the goal of the attack?

- the paper is not very well structured which makes it hard to read, and many important information are scattered throughout the paper. this work needs to be better presented so that the reader is not confused about the proposed method. For instance, in Eq1, V and J_R are present but they are not introduced until much later in the paper. The same as for Band C (budget limits) introduced in Eq.2. Alg2 seems to be essential and is referenced several times in the main paper, but its definition is in the appendix and it is not explained exactly what it does, when the reader needs it. Please fix the structuring issues, and improve presentation.

- In general, the proposed method is closely related to [ref1], which limits its novelty and contributions. Please discuss how this work is different than [ref1].

**Questions:**

see weaknesses.

---

### Note · Authors · 2024-12-04

**Comment:**

We appreciate the insightful comments from the reviewers and will incorporate them in the next version of our work

**Withdrawal Confirmation:**

I have read and agree with the venue's withdrawal policy on behalf of myself and my co-authors.